# Serotonin in Animal Cognition and Behavior

**DOI:** 10.3390/ijms21051649

**Published:** 2020-02-28

**Authors:** Julien Bacqué-Cazenave, Rahul Bharatiya, Grégory Barrière, Jean-Paul Delbecque, Nouhaila Bouguiyoud, Giuseppe Di Giovanni, Daniel Cattaert, Philippe De Deurwaerdère

**Affiliations:** 1INCIA, UMR5287, Centre National de la Recherche Scientifique, 33076 Bordeaux, France; julien.bacque-cazenave@u-bordeaux.fr (J.B.-C.); bhartiyarahul20@gmail.com (R.B.); gregory.barriere@u-bordeaux.fr (G.B.); jean-paul.delbecque@u-bordeaux.fr (J.-P.D.); nbouguiyoud96@gmail.com (N.B.); 2Department of Biomedical Sciences, Section of Neuroscience and Clinical Pharmacology, University of Cagliari, 09100 Cagliari, Italy; 3Laboratory of Neurophysiology, Department of Physiology and Biochemistry, Faculty of Medicine and Surgery, University of Malta, MSD 2080 Msida, Malta; giuseppe.digiovanni@um.edu.mt; 4School of Biosciences, Neuroscience Division, Cardiff University, Cardiff CF24 4HQ, UK

**Keywords:** serotonin receptor, animal phyla, mood, decision-making, locomotion, anxiety, neuronal excitability, feeding, aggressiveness, impulsive/compulsive dimension

## Abstract

Serotonin (5-hydroxytryptamine, 5-HT) is acknowledged as a major neuromodulator of nervous systems in both invertebrates and vertebrates. It has been proposed for several decades that it impacts animal cognition and behavior. In spite of a completely distinct organization of the 5-HT systems across the animal kingdom, several lines of evidence suggest that the influences of 5-HT on behavior and cognition are evolutionary conserved. In this review, we have selected some behaviors classically evoked when addressing the roles of 5-HT on nervous system functions. In particular, we focus on the motor activity, arousal, sleep and circadian rhythm, feeding, social interactions and aggressiveness, anxiety, mood, learning and memory, or impulsive/compulsive dimension and behavioral flexibility. The roles of 5-HT, illustrated in both invertebrates and vertebrates, show that it is more able to potentiate or mitigate the neuronal responses necessary for the fine-tuning of most behaviors, rather than to trigger or halt a specific behavior. 5-HT is, therefore, the prototypical neuromodulator fundamentally involved in the adaptation of all organisms across the animal kingdom.

## 1. Introduction

Serotonin (5-hydroxytryptamine, 5-HT) is a neuromodulator present in most animal phyla, although the overall function of this small molecule is still poorly understood. As cited by Azmitia and Jacobs in 1992, “although 5-HT has been implicated in a wide range of physiological and behavioral processes in vertebrates, it does not appear to be essential for any of them” [1]. This definition of the role of a neuromodulator on the operations of the central nervous system (CNS) is perfectly adapted in the case of 5-HT systems compared to other neuromodulatory systems such as catecholamines, acetylcholine, histamine, and neuropeptides. 

Either in vertebrates or invertebrates, the 5-HT systems have been shown to modulate cognitive and behavioral functions. Often referred to as the “happiness hormone”, in part due to the positive effects of selective 5-HT reuptake inhibitors (SSRIs) in some patients with depressive disorders, the role of 5-HT is much broader and more complex because it is involved in stress responses and mood disorders at different levels [2,3,4]. The diversity of modulating effects induced by 5-HT in cognitive tasks and behavioral responses is linked to *i)* its simultaneous effects on a multiplicity of neural targets underlying these functions and *ii)* to the large number of its receptors with their intracellular signaling pathways and their different affinities, acting at various neuron locations. Because of these specificities, 5-HT systems can ensure fine-tuning of behaviors in various situations, sometimes by inhibiting learned behavioral responses that would be inappropriate or by adjusting the timing of responses to ensure more adapted behavior [5,6]. We have explored the influence of the 5-HT system in some biological functions in both vertebrates and invertebrates. We have covered specific animal responses to illustrate our topic as reported in Figure 1.

## 2. Organization of 5-HT Systems in Animals

The organization of the 5-HT systems is completely different between species ranging from a limited number of cells in *Aplysia* or *Drosophila* (100 5-HT immunoreactive cells) to several thousand neurons in vertebrates. The variety and the heterogeneity of 5-HT neurons have been highlighted in vertebrates [7] as well as in invertebrates [8]. In general, the 5-HT systems in animals comprise 5-HT neurons, which share a particular shape made of thousands of varicosities, leading to the concept of volume transmission [9], and several 5-HT receptors (5-HTRs). In vertebrates, 5-HT neurons are localized in the midbrain raphe nuclei described as the B1–B9 cell groups [1,10]. The raphe pallidus, obscurus, and pontis form a caudal cluster (B1–B5), whereas the dorsal raphe and median raphe form a rostral cluster in the pons (B6–B9) [11]. The caudal raphe group projects to the spinal cord and the rostral group to the forebrain via an extensive and diffuse innervation. The dorsal raphe nucleus contains the highest number of 5-HT neurons and its anatomical sub-regions display some degrees of specific innervation of the forebrain [12,13,14]. The median and dorsal raphe nuclei receive excitatory and inhibitory inputs from most brain areas [15]. In crustaceans or insects, 5-HT neurons are widely present in each of the ventral cord ganglia, display large local ramifications that act in multiple neuropil areas, and some axonal branches form three pairs of rostrocaudal fibers [16,17,18,19]. Presumably, two of those fibers would project anteriorly and one posteriorly to the entire nervous system [20]. 5-HT neurons are often localized close to sensory integration input area in the brain of arthropods and some 5-HT cells in the abdominal ganglia of crayfish nerve cord are sensitive to the mechanical stimulation of abdominal segmental fringe hairs [21]. In cnidarians, 5-HT cells are also close to the sensory organs [22]. 5-HT is released from the terminals or varicosities, while its course can be prolonged away from the releasing sites (extrasynaptically). The specificity of 5-HT transmission also comes from reuptake sites, which can be highly specific (serotonin transporter, SERT) or less specific including catecholaminergic transporters [23]. Thus various modalities of neurotransmission have been reported in the case of 5-HT including classical synapses, neuro-humoral, and/or paracrine influences [9]. The latter has been well shown in an insect, the female cricket *Acheta domestica*, where 5-HT innervates the genital chamber muscles without identified synapses [24]. These modalities are more or less present across the animal kingdom. 

The metabolic pathways of 5-HT also share similar features between species. The amino acid precursor tryptophan is hydroxylated by tryptophan hydroxylase, which presents two isoforms in various species [25,26,27]. The product 5-hydroxytryptophan (5-HTP) is decarboxylated by the aromatic amino acid decarboxylase into 5-HT. 5-HT can reach the vesicular compartments through the vesicular transporter for monoamines. The catabolism of 5-HT is complicated, not so by the recruited enzymatic pathways involving the monoamine oxidase A or B (MAOA, MAOB), and aldehyde dehydrogenase to produce the final metabolite 5-hydroxyindole acetic acid (5-HIAA) [28,29]. The catabolism is complicated because it preferentially involves MAOA enzyme in mammals and this subtype, at variance with MAOB, is usually not located at terminals of 5-HT neurons [30]. It would imply that the catabolism of 5-HT is in part associated with other cells. In any case, 5-HT and its metabolite 5-HIAA are found in many different species. The tissue quantities of 5-HT follow in general the density of fibers and cell bodies. In crayfish, the highest quantity is found in the cerebroid ganglia (approximately 300 pg/mg), followed by the thoracic chain (120 pg/mg) and the abdominal chain (10 pg/mg) [31,32]. In mammals, the highest quantities of 5-HT are found in the substantia nigra (500–1000 pg/mg), followed by ventral tegmental area and raphe nuclei. Moderate to high quantities are found in other brain regions and spinal cord [33,34,35,36,37]. In chicken, the 5-HT content is higher in the amygdala compared to the thalamus or the striatum [38]. The index of the turnover (5-HIAA/5-HT) is lower in chicken compared to mammals [34,38]. In Indo-Pacific Bluestreak cleaner wrasse, *Labroides dimidiatus*, the 5-HT content is quite low (approximately 20 pg/mg) and homogenous in the forebrain, the diencephalon, and the cerebellum of the fish [39]. Of note, the quantities of 5-HT can evolve according to the status of the organism. In the mollusk *Perna perna*, the quantities of 5-HT are quite low at the resting stage of the sexual cycle (50, 70, and 8 pg/mg in the cerebroid ganglia, the pedal ganglia, and the gonads, respectively), but these contents are increased at the final stage of sexual maturation and egg-laying (300, 140, and 17 pg/mg, respectively) [40]. 

5-HT acts on a variety of receptors. In mammals, 5-HTRs are organized in seven families (5-HT1-7R) and more than 16 subtypes have been described. They have been characterized for several years regarding cloning, pharmacology, and heterogeneous CNS expression [41]. Conversely, 5-HTRs are less known in invertebrates despite the prediction that they could be almost similar [42]. Briefly, four 5-HTRs named 5-HT_1Adro_ (or d5-HT_1A_), 5-HT_1Bdro_, 5-HT_2dro_, and 5-HT_7droR_ have been discovered in *Drosophila* and three in crustaceans (5-HT_1Acrust_, 5-HT_2Bcrust_, and 5HT_7crust_) [19,43,44,45,46]. In mollusks, two 5-HTRs have been cloned in *Lymnaea stagnalis* belonging to 5-HT_1_R and 5-HT_2_R families [47,48], whereas five have been cloned in *Aplysia* [49]: 5-HT_ap1_ and 5-HT_ap2_ inhibited adenylate cyclase activity and 5-HT_apAC1_ stimulated it [49]. In a pond snail (*Helisoma trivolvis*), two receptors types (5-HT_1Hel_ and 5-HT_7Hel_) have been cloned and characterized [50]. In crayfish, 5-HTRs are widely distributed in various nervous structures with a specific localization in each one. For example, 5-HT_1crust_ are localized in thoracic ganglia surrounding axonal branches projecting into the second nerves of each ganglion and more specifically close to the neuropil part of motoneurons [51]. This receptor may be localized in the same area where 5-HT fibers form many close appositions with motoneurons and are involved in classic inhibitory synaptic responses. Conversely, 5-HT_2crust_ are localized more centrally in the crayfish nervous ganglia where 5-HT fibers are not co-localized with axon of some motoneurons and may be involved in a more global excitatory paracrine response. These receptors would modulate the membrane potential act via an action involving K^+^ channels) [52]. 

The pharmacology of 5-HT receptors is complex, and it is clear now that responses to 5-HT drugs differ between invertebrate and vertebrate receptors. Moreover, pharmacological responses of one 5-HT receptor inside the same phylum could be different [53]. For example, 5-methoxy-tryptamine, a 5-HT_1_R agonist in vertebrates, has variable effects on invertebrate receptors, triggering some activities from 5-HT1Rs in honeybee [54] and beetle [55], but low or no activity in *Manduca sexta* and cockroach [56]. 

The combination of all these factors implies that the 5-HT system exerts complex regulations from the cellular level, notably by controlling cell and neuronal excitability to the entire organism [57].

## 3. Motor activity and Locomotion

5-HT is a major neuromodulator of motor behaviors (Figure 1A) in many species belonging to different invertebrate phyla. This includes flatworms [58], cnidarians [59], nematodes [60], annelids [61,62], arthropods (insects and crustaceans, [63,64,65]), and mollusks [66,67,68,69]. Part of our knowledge comes from studies investigating the ecological/toxicological consequences of SSRIs used as antidepressants, released in sanitary water, and finally spread in rivers [63,65,70,71,72]. It also comes from studies stimulated by the need to develop new ecological intervention to decrease or eradicate associated pathologies to promote human health [58,73].

5-HT has been shown to be involved in different aspects of the locomotor functions. In the nematode *Caenorhabditis elegans*, 5-HT contributes to the selection of the locomotor mode as it allows for the behavioral transition from crawling to swimming [74] and modulates the level of locomotor activity and the regulation of locomotor speed [75]. Notably, it allows the slowing down of locomotor exploration when an unfed animal encounters food [76], an action mediated by the Mod-1 5-HT gated chloride channel and the SER-4R [60,77]. An optogenetic approach in *C. elegans* has indicated that 5-HT is also involved in the control of the direction of locomotor behavior, the inhibition of the activity of 5-HT cells triggering backward locomotion [78]. 

In *Drosophila* larvae, increasing the level of 5-HT is able to decrease body wall contractions (locomotion). In the larval stages, 5-HT is involved in turning behavior [79], whereas in adult flies, acute activation of 5-HT neurons disrupts normal locomotor activity [64]. In light of the absence of 5-HT at the neuromuscular junction [80], such an effect implies an important regulatory role of the central 5-HT systems over the motor programs. This has been further evaluated by using mutations in the 5-HT biosynthetic enzymes and by knocking down 5-HTRs expressed in brain mushroom bodies (MBs) (d5-HT_1A_, d5-HT_1B_, d5-HT_2_, and d5-HT_7_) using RNAi strategy. Animals expressing RNAi for d5-HT_1B_, d5-HT_2_ or d5-HT_7_, but not d5-HT_1A_ in MBs showed increased motor behavior. Thus, all three d5-HTRs expressed in MBs decrease locomotor activity in fly larvae, although the activation of d5-HT_1B_ in other CNS regions may counteract its own action in the MBs [81]. Similarly, it has been reported in the leech that 5-HT activates the locomotor network at the nerve cord level, but inhibits locomotion at the head brain level [61]. These observations illustrate the complexity of the serotonergic modulations of locomotor networks. It is also worth mentioning that 5-HT action may differ between closely related species as it has been described in nudibranch mollusks [67,68].

Central 5-HT is also a powerful neuromodulator of locomotor activities in various vertebrates, including lamprey [82], zebrafish [83,84,85], *Xenopus* [86,87], and rodents (mouse, rat, rabbit; [88,89,90]. In spinal rats and mice, serotonergic agonists are amongst the most powerful drugs promoting the activation of the sub-lesional locomotor networks and the recovery of locomotor movements following a spinal cord injury [91,92,93]. This strong modulatory action is actually observed as early as the very first post-natal days in rodents. The stimulation of spinal cord projecting serotonergic neurons or exogenous application of 5-HTR agonists on ex vivo spinal cord from newborn rats initiates and sustains episodes of so-called fictive locomotion [89,94,95,96,97,98].

5-HT also influences motor activities in vertebrates [99,100] and invertebrates by directly acting on the motoneurons that drive motor contractions. 5-HT enhances the motoneuron drive in *Drosophila* larvae [101]. In leeches, 5-HT modulates the membrane properties of excitatory swim motor neurons [102]. In crustacean motoneurons, 5-HT increases tonic discharge at 10 µM, but decreases it at 100 µM. These two responses involve different types of receptors: The excitatory effect involves 5-HTRs localized in the neurites that act by closing a K^+^ conductance; the inhibitory response involves 5-HTRs receptors located on the axon close to the action potential initiation segment acting by opening a K^+^ conductance [52]. Recently, it has been demonstrated that this duality of 5-HT effects on crayfish motoneurons can be achieved simultaneously in different leg motor neurons by the activation of a single 5-HT cell [103]. These facilitatory and inhibitory states coexist, and they are mediated by cAMP/protein kinase A signaling pathways. These states compete and the emergence of one promotes the decay of the other [104].

Finally, in some invertebrates, 5-HT can peripherally exert its motor action, where it modulates the efficacy of the neuromuscular junctions and regulates muscle contraction (for review see [80]; *C. elegans*: [105]). In crustaceans, 5-HT enhances synaptic transmission [106] at the level of the neuromuscular junction by a presynaptic action [107]. 5-HT also facilitates the contraction of the buccal muscle at the neuromuscular junction level in *Aplysia* [108] where it enhances the presynaptic transmitter release [109]. In contrast, the release of 5-HT by Retzius cells in the muscles of the leech decreases the presynaptic release of acetylcholine resulting in a decrease in muscle contraction.

## 4. Arousal 

Several studies have pointed out the relationship between 5-HT systems and arousal/activity [1,5]. In particular, the significant activation of a group of 5-HT neurons has been associated with activities such as grooming, oral activity, chewing, or biting. The activation of these neurons is also observed when the face or the head are stimulated [110]. Repetitive motor activities also seem to be able to activate 5-HT neurons [107]. Finally, although stressful situations do not produce a systematic change in the activity of 5-HT neurons, chemosensitive 5-HT neurons respond to acidosis in several raphe nuclei by increasing their frequency of discharge [111].

The links between arousal and 5-HT systems have been studied in Gastropods and in particular in *Aplysia*, in which two main classes of arousal elements have been demonstrated: General versus localized [112]. The localized arousal elements can mediate specific arousal effects, often restricted to a single class of behavior. The general arousal elements produce effects less specific of arousal and can influence several classes of behavior. The modulating effects of 5-HT can involve both classes of elements.

In *Drosophila*, the role of 5-HT is not as clear as in Gastropods, and dopamine could also be involved in arousal [113].

## 5. Sleep and Circadian Rhythm

In mammals, 5-HT (Figure 1B) is implicated in sleep-wake states [114]. In cats and rodents, 5-HT seems to promote wakefulness and prevents the sleep phase with rapid eye movements (REM), likely via 5-HT_1_R. Indeed, mutant mice that do not express 5-HT_1A_ or 5-HT_1B_R exhibit greater amounts of REMS than their wild-type counterparts. REM sleep is associated with an atonic state of skeletal muscles. The discharge of 5-HT neurons is generally low, and it is still reduced during slow-wave sleep [111]. The participation of 5-HT in the control of the circadian rhythm is supported by the fact that there is a significant projection of 5-HT neurons from the raphe nuclei to the suprachiasmatic nucleus, a tiny region of the hypothalamic nucleus, which regulates the sleep-wake cycle [111]. This region contains the circadian clock in pigeons, and 5-HT has been shown to be involved in its synchronization in pigeons [115]. In addition, in doves, 5-HT levels in serum seem to be positively correlated with the circadian activity rhythm [116]. Finally, one of the metabolites of 5-HT is melatonin, a molecule that is known to regulate the sleep–wake cycle.

As is often the case with 5-HT, opposite effects can be obtained depending on the type of receptor considered. Thus, in mice, the 5-HT_1A_ and 5-HT_1B_R reduce REM sleep, while 5-HT_2A_R, 5-HT_2C_R or 5-HT_7_R produce the opposite effect [111]. These effects imply different mechanisms acting on different targets.

The relationship between circadian rhythm and sleep-wake has been also studied. In vertebrates, 5-HT seems to inhibit the effects of light on the circadian activity [107]. In *Drosophila*, this effect of light on circadian rhythm is mediated by the degradation of the clock protein *timeless* induced by light, and 5-HT prevents this degradation via the activation of d5-HT_1B_R [117]. In addition, the *Drosophila* 5-HT_2_R (d5-HT_2_) seems to play a role in the circadian activity of stage 3 larvae, by controlling their anticipatory behavior [118]. In *Drosophila*, playing on the 5-HT system can break the link between sleep and the circadian rhythm. In fact, an increase in 5-HT reinforces sleep, presumably via the *Drosophila* 5-HT_1A_R (d5-HT_1A_) while the deletion of tryptophan hydroxylase suppresses sleep at night [119]. Likewise, the sleep of a fly, in which d5-HT_1A_ has been suppressed, is reduced and fragmented, while the re-expression of d5-HT_1A_ in MBs saves the normal phenotype [120].

In crustaceans, the situation is much less known compared to *Drosophila* [121]. Regulation of circadian rhythm is operated in both the central brain and the eyestalk X-organ-sinus gland system, where 5-HT and melatonin have been largely found [121]. In crayfish, 5-HT concentration in tissue displays circadian fluctuation, linking the 5-HT system with the pacemaker system, involved in circadian rhythm [122]. Moreover the concentration of 5-HTR_1crust_ in the eyestalks is lower during the dark phase that the light phase [123]. Similarly, the expression of 5-HT_7crust_ in thoracic ganglia is increased during a constant light exposition [46].

## 6. Feeding 

The control of feeding behavior by 5-HT (Figure 1C) has been extensively studied [124]. In humans, obesity treatments have largely used 5-HT-based pharmacological agents and notably d-fenfluramine. In mice, d-fenfluramine enhances the release of 5-HT, which could secondarily activate 5-HT_2C_Rs leading to its anorectic effects. However, due to its toxic effects on the heart, d-fenfluramine has been withdrawn from the market, and it has been replaced by the 5-HT_2C_R agonist lorcaserin [125]. The multiplicity of mechanisms by which 5-HT modulates feeding behavior is attested by the variety of 5-HTRs implicated (5-HT_1A_R, 5-HT_1B_R, 5-HT_2C_R, 5-HT_4_R), and by the diversity of the brain areas involved in this behavior, for example sub-regions of the hypothalamus [126], the ventral tegmental area, or the nucleus of the solitary tract [127,128]. 5-HT would inhibit feeding behavior via reinforcement of satiety signals and the extension of their duration, likely via a diversity of 5-HT receptors. Thereby, the size of meals is reduced [121]. Some effects of 5-HT are also indirect, acting on dopamine systems via 5-HT_2C_R in cortical and subcortical areas [129,130,131] to reduce the reinforcing features of food and/or to impede impulsive behavior targeting food [132,133]. The stimulation of 5-HT_4_R may also inhibit feeding behavior in rodents, likely via its action at the level of the nucleus accumbens [134,135]. Its higher expression in the nucleus accumbens and ventral pallidum has been associated with a higher body mass index in humans [136,137].

Also, in invertebrates, 5-HT controls food behavior, for example in *C. elegans* [138], in which 5-HT activates the pumping behavior via the 5-HTR SER-7. This pumping is used by *C. elegans* to fill its gut with the bacteria present in its environment. By using a microfluidic system to precisely control the concentration of available food and to monitor the dynamics of pharyngeal pumping at high resolution, it has been shown that feeding is characterized by rapid and regular pumping bursts, the duration and frequency of which are correlated with food availability. However, when the food density is low, long breaks interrupt eating behavior. Interestingly, 5-HT promotes rapid pumping and eliminates long breaks. Moreover, the synthesis of 5-HT in pharyngeal secretory neurons (NSM) and a pair of 5-HT motor neurons located in the middle of the worm body (HSN) are necessary and sufficient to control fast pumping. Two distinct mechanisms control fast pumping: One via the 5-HT_1_R ortholog SER-1 is involved in food-dependent induction of fast pumping (the 5-HT_2_R ortholog SER-4 could also be involved in maintaining the high pumping rate); the other is involved in feeding activated by exogenous 5-HT via the 5-HT_7_R ortholog SER-7 and SER-4 [139].

Also, in insects, 5-HT is involved in the control of feeding behavior [140]. For example, in *Rhodnius proxilus*, the rapid production of urine within minutes of feeding prevents blood meal to compromise salt and water balance. The activity of Malpighian tubules producing urine is controlled by a range of neuropeptide families and serotonin acting upon a variety of tissues. In *Drosophila*, disruption of components the 5-HT system significantly impairs locomotion and feeding behaviors in larvae [61]. For example, d5-HT_1B_R and d5-HT_2A_R knockdowns significantly decrease mouth hook movements in the larvae. Inversely, chronic activation of 5-HT neurons by TrpA1 channels increases feeding behavior.

In the Chinese mitten crab (*Eriocheir sinensis*), the 5-HT_7_R (5-HT_7_Crust receptor) was characterized. The expression levels of this receptor increase significantly after feeding and could be involved in the physiological regulation of the hepatopancreas and intestines after ingestion [43].

## 7. Social Interactions, Social Status, and/or Aggressiveness

Numerous studies have shown that 5-HT participates in the control of social interactions and status in invertebrates and vertebrates (Figure 1D). Repeated fights between animals are usually addressed to study these social interactions and lead to establishing the status of dominant (winner of more and more fights) and subordinate (loser of more and more fights) [141,142,143]. After 15 days of crayfish pairings, the sensitivity to 5-HT application of giant fibers involving in the escape behavior circuits evolves. 5-HT improves the response to mechanosensory inputs in isolate and dominant crayfish, whereas the same 5-HT application has a weak or no effect in the subordinate animal [144]. The change of sensitivity to 5-HT is gradual during the 15 days of pairing and is associated with modifications of body posture and the neurobiological network underlying escape behavior [145] or locomotion [146]. In addition, 5-HT could play a role in the reversal of the behavior and the issue of the fight. Injection of 5-HT can considerably renew the willingness of the slightly lowest physical size crayfish to engage the dominant in further agonistic encounters [147]. Moreover, 5-HT elicits opposite behavioral responses in the larger rival injected [148]. These results suggest that the effects of 5-HT on aggressiveness depend on the perceived relative size difference of the opponent. It further supports the idea that 5-HT does not directly act on aggressiveness but rather on the brain centers integrating risk assessment. Moreover, it implies that 5-HTmodifies several behavioral parameters interfering with the aggressiveness, such as the likelihood of retreat, the duration of fighting, or again, fight initiation. Different studies show that the metabolic pathway of 5-HT modulates this behavioral reversal. Indeed, the precursor of 5-HT, 5-HTP, reduced aggression in pigeons without affecting feeding in dominant members in the context of food competition, and decreased the time spent to flee by subordinates [149]. In the reptile *Anolis carolinensis*, it has been also demonstrated that the chronic exposure to the SSRI sertraline reverses the dominant social status likely via an enhancement of extracellular 5-HT levels [150].

The studies performed in insects give similar doubts with regard to the effect of 5-HT. In crickets, the enhancement of aggressiveness is mediated by octopamine, whereas 5-HT promotes escape behavior [142]. However, hyperactivity is frequently observed after the depletion of 5-HT levels without affecting the level of aggressiveness. After a fight, losers exhibit escape behavior associated with lower levels of brain 5-HT compared to winner crickets whose 5-HT levels remained identical [151]. Similarly, in *Drosophila*, the level of aggression in fruit flies is not altered by the increase or the decrease in 5-HT synthesis. Nevertheless, the fighting ability is reduced by the disruption of 5-HT transmission whereas an increase in 5-HT synthesis or activity of 5-HT neurons elicits aggression [152,153]. In *Drosophila,* a single 5-HT neuron expressing a d5-HT_1A_R would drive aggressive behavior [8]. This specific neuron is part of a small group of 5-HT neurons modulating the regulation of aggression via some GABAergic and cholinergic neurons receiving sensory inputs from a well-known integration center in the brain of *Drosophilia*. These two types of neurons are involved in aggressiveness behavior and express 5-HT_1A_Rs; they respectively lead to a decrease in GABA release or a decrease in cholinergic neurons activity [154]. Recent findings in insect models have permitted to elaborate on the 5-HT mechanisms involved in the regulation of the circuitry of aggression in invertebrate [155]. 

Studies recently performed in *Octopus bimaculoides* confirmed the involvement of 5-HT neurotransmission in regulating social behaviors [156]. There was a dramatic improvement in the prosocial behavior of *O. bimaculoides* after (+/)-3,4-methylenedioxymethamphetamine (MDMA) exposure. During the pre-trial session, subject animals spent more time in the object chamber compared to the center chamber, whereas in the MDMA-induced post-trial, subject animals spent significantly more time in the social chamber compared to the center chamber. The genome sequencing showed that the sixth transmembrane domain of the SERT encoded by the SLC6A4 gene was the principle binding site of MDMA as well as 5-HT. The SLC6A4 gene in the octopus orthologs within this region is 100% identical and aligned to the human SLC6A4 gene [156].

Numerous studies have evoked the concept of mutualistic cooperation in the world of fish animals in cleaner/client fish case, a cooperation of cleaning from ectoparasites applied by the cleaners on clients [157]. Studies performed on client reef fish adult blond tang *Naso elegans* showed neurochemical changes of the 5-HT system in the cooperative behavior. There was an increase in the level of 5-HT’s metabolite 5-HIAA in two brain areas, the forebrain and the diencephalon, when interacting with cleaners rather than conspecifics [158]. On the other hand, studies performed in the Indo-Pacific Bluestreak cleaner wrasse *Labroides dimidiatus* fish showed a decreased forebrain 5-HT levels in the brain of cleaners interacting with clients, but an increase in diencephalon 5-HT levels when the cleaners were introduced to clients inside another aquarium [39]. The study performed in *L. dimidiatus* fish showed that the SSRI fluoxetine and the 5-HT_1A_R agonist 8-OH-DPAT enhanced the proportion of inspection of clients by the cleaners. Opposite results were obtained with the 5-HT_1A_R antagonist WAY 100,635 and the tryptophan hydroxylase inhibitor p-chlorophenylalanine. 8-OH-DPAT and WAY 100,635 increased and decreased the duration spent inspecting clients and the client’s body jolts (kind of a response to cleaner’s bites on clients), respectively [157]. 

The link between social status or aggression and 5-HT has been extensively studied in mammals and notably mice [159]. This field of research is complicated because aggressive behaviors and social interactions can be investigated through different ways and parameters, and are dependent on the strain [159]. A higher level of aggressiveness has been associated in several studies with lower tissue levels of 5-HT, but it is still difficult to generalize [160]. Low levels of 5-HIAA in the CSF in macaques coincided with the anti-social behavior marked by aggression and violence [161]. In mice or healthy volunteers, it has been reported that tryptophan-free diet increases the level of aggressiveness although it occurred under specific circumstances [162,163]. 

The aggressive/anti-aggressive responses are mediated by different 5-HTRs. The 5-HT_1A/IB_R agonists eltoprazine or TFMPP, and the preferential 5-HT_1B_R agonists zolmitriptan, anpirtoline, and CP-94,253 reduce numerous aspects of aggressiveness in mice and/or rats. Conversely, mice lacking 5-HT_1B_R display higher rates of aggressive episodes [143,164,165,166]. The neuropharmacological basis underlying the efficacy of 5-HT_1A/1B_R agonists is unclear. It could be indirectly mediated by the inhibition of 5-HT neuron activity consequent to the activation of 5-HT_1A_ and 5-HT_1B_ autoreceptors and/or the activation of 5-HT_1A_R and 5-HT_1B_R expressed by 5-HT receptive cells in diverse neurobiological networks [160,164]. The plurality of both the neurobiological networks and the 5-HTRs involved in aggressive behavior has been well exemplified in *Drosophila*. Indeed, the level of aggression of flies was increased by the 5-HT_1A_R agonist 8-OHDPAT, mainly wing threats and fencing components, whereas the aggressiveness corresponding to lunging and boxing components was decreased by the 5-HT_2_R agonist DOI [167]. A similar distinction of 5-HTRs involvement has been reported in male–male aggression of the cricket *Gryllus bimaculatus* during courtship behavior. The 5-HT_1_ and 5-HT_7_Rs mediate male-male aggression consequent to prior female contact whereas 5-HT_2_R maintains a low level of aggressiveness after social defeat [168]. 

It is still difficult to attribute the abovementioned mechanisms engaged by the 5-HT system to a specific control of aggressive behavior because the neurobiological networks recruited in the aggressive responses are likely numerous beyond the intrinsic complexity of aggressive behavior. An action of 5-HT on other behaviors indirectly modulating the decision to flee or to fight with an opponent cannot always be discarded in contrast to an indirect action of 5-HT on motor activity, which can be tested. For instance, the anti-aggressive effect of eltoprazine in mice could be sustained by its anxiogenic properties [165]. 

## 8. Anxiety

Anxiety is a complex response to stress in animal behavior, which can take several forms [169]. Anxiety and anxious states have been extensively investigated in numerous clinical and preclinical studies in mammals and the link with the 5-HT (Figure 1E) is acknowledged by the numerous marketed anxiolytic drugs having a 5-HT profile. Anxiety occurs when the stressor is absent or not clearly identified and is therefore considered a secondary emotion. Anxiety-like behavior has been reported in invertebrates, particularly in crayfish and *Drosophila*. Physical or social stress imposed to crayfish lead to avoidance responses to light exposure. This type of responses of crayfish shares similarities with rodent behavior in some behavioral tests including the plus maze [32,170]. The avoidance response in crayfish was associated with an increase of 5-HT concentrations in the brain. Furthermore, the administration of exogenous 5-HT can trigger avoidance response whereas a cocktail of 5-HTR antagonists as well as by the anxiolytic chlordiazepoxide used as comparative drugs reduced anxiety-like behavior of crayfish [32]. Interestingly, dopamine would have no role in stress response and anxiety-like behavior, highlighting a specific role of 5-HT in promoting this type of behavior [31]. *Drosophila* submitted to stress can also exhibit anxiety-like behavior that shares similarities with rodent behavior; it is reduced by the benzodiazepine and anxiolytic drug diazepam [171]. The anxiety-like behavior involves 5-HT system and is regulated by d5-HT_1B_R, and d-5-HT_2B_R but not 5-HT_1A_R and d5-HT_2A_R. Thus, the deletion of d5-HT_1B_R or the overexpression of the dSERT respectively increased and decreased anxiety-like behavior. Conversely, the deletion or the impairment of d-5-HT_2B_R diminished anxiety-like parameters [171]. 

In rodents, several 5-HTRs including 5-HT_1A_R, 5-HT_1B_R, 5-HT_2A_R, 5-HT_2B_R, 5-HT_2C_R, 5-HT_3_R, 5-HT_4_R, and 5-HT_7_Rs participate in anxiety-like responses and both agonists and antagonists can be used depending on the subtype to regulate anxiety-like behaviors [172]. This multiplicity of 5-HTRs modulating anxiety is not surprising because anxious states involve multiple brain regions organized in numerous, intermingled neurobiological networks [169]. Some of the 5-HTRs may alter, in an opposite manner, anxious states and it is more difficult to determine whether an increase or a decrease in 5-HT transmission promotes anxiety. Finally, even the same receptor subtype can trigger distinct responses on anxious states. 5-HT_2C_R agonists, which act on monoaminergic transmission in various brain regions [33,131], can display no effect or trigger anxiogenic or panicolytic profiles depending on the laboratory tests used and the brain location of action [173]. Conversely, 5-HT_2C_R antagonists exhibit an anxiolytic profile in several but not all behavioral paradigms [173,174]. Nowadays, SSRI or partial 5-HT_1A_R agonists are the 5-HT agents that are prescribed for the treatment of anxious states. A better understanding of the role of 5-HTRs in anxiety could ultimately lead to a targeted pharmacological approach according to the nature of the anxious states. 

## 9. Mood

Mood is a connoted word that is not specific to an emotional status and is associated with a relatively long-lasting emotional status. Mood disorders typically refer to the various forms of depression and bipolar disorders. Mood alteration can be also studied in invertebrates using specific conditions (Figure 1F). It has been reported that crayfish exhibit an increase of anxiety-like behavior upon repeated defeats with a stronger congener during fights. The level of anxiety of harassed crayfish (loser) by the winner is directly related to the harassment period duration [170]. The loser will progressively present a more pronounced subordinate behavior when this social stress is prolonged for several days or weeks, marked by a progressive decrease in its ability to react. It could correspond to depression-like behavior. The 5-HT is likely involved in this behavior mainly at the level of its regulatory action on cells and networks. Indeed, the giant fiber escape circuit [144,145] and the locomotor network [146] were no longer responding to 5-HT modulatory effects in subordinate crayfish. The loss of behavioral responses in these animals is clearly associated with a clear depression of neural activity. In *Drosophila*, chronic stress induced by vibration stress over three days also resulted in drastic changes of behavior associated with the loss of 5-HT neuromodulation [175]. This depression-like state controlled by 5-HT signaling in alpha and gamma lobes of the MBs is ameliorated by lithium chloride exposure. The authors reported reduced 5-HT release at MBs in chronically stressed *Drosophila*. The 5-hydroxytryptophan or sucrose reversed the behavioral changes and elevated 5-HT levels in the brain acting at d5-HT_1A_R.

These recent findings in *Drosophila* and crayfish remind us the reported role of the 5-HT system in depressive symptoms observed in mammals and humans in particular. In humans presenting with major depressive disorders associated with suicide tentative, reductions of 5-HT levels in CSF or the brain have been occasionally found. Moreover, a diet without tryptophan could promote depressive status in individuals vulnerable to mood disorders. However, a direct relationship between 5-HT levels and depressive states does not exist in mammals [2]. The 5-HT systems play an essential role in the mechanism of action of antidepressant drugs whatever their pharmacological action in interaction with the other monoaminergic systems [4,176,177,178,179]. The loss of reactivity of neurobiological networks to 5-HT as reported in crayfish is worth considering and it is probably true for other neurotransmitters. The positive action of antidepressant drugs could be related to remodeling of 5-HT transmission in different parts of the brain enabling a better excitability of the neurobiological networks. In any case, their action takes time, which is still the main concern in the clinic, is probably widespread, and the efficacy is not guaranteed [4,30,180]. 

## 10. Learning and Memory

5-HT system has been involved for a long time in learning and memory (Figure 1G). Invertebrates have been successfully investigated in some elementary forms of behavioral learning associated with changes of reactivity of the 5-HT system. The neuromuscular synapses in buccal motoneurons in *Aplysia* are one of the first evidence of adapted behavioral responses related to changes of synaptic strength. Indeed, an increase in the release of glutamate at sensory–motor neuron synapses sustained by 5-HT has been determined in the persistent facilitation of the withdrawal reflex. A 5-HTR coupled to adenylate cyclase has been involved in this response [49]. Short- and long-term plasticities in invertebrates have also been reported, notably in *Drosophila* and honeybee, *Apis mellifera* [181]. The findings originally described in invertebrates regarding the long-lasting plasticity and the influence of the 5-HT system has been found in vertebrates with some analogy [182]. The 5-HT system in learning and memory mobilizes the majority of 5-HTR subtypes that play a role in behaviors that involve a high cognitive demand and in memory improvement [183]. The precise influence of 5-HTR subtypes deserves additional studies in view of their possible targeting in the clinic as cognitive enhancer [184]. It has been recently stressed out for 5-HT_4_R and 5-HT_6_R subtypes [137,185]. Several 5-HT_6_R antagonists have been tested in recent clinical trials in Alzheimer patients although the results were mitigated [186]. Nonetheless, this field of research is complicated because, as mentioned above for other behavioral responses, 5-HTRs can act in the opposite manner, often in a different manner, in memory processes [183,184,187]. An additional complexity in learning and memory processes is that some learning responses such as the conditioned eyeblink response in rabbits or the auto-shaping learning task in rats are independent of the levels of 5-HT [188,189]. These responses are sensitive to some inverse agonists suggesting that the controls exerted by some 5-HTRs over learning processes can involve the constitutive activity of some 5-HTRs including 5-HT_2A_R and 5-HT_2B/2C_R [190,191,192].

## 11. Impulsive/Compulsive Dimension and Behavioral Flexibility

Impulsivity, compulsivity, and behavioral flexibility participate in the personality traits of individuals and decision-making. In humans, maladaptive decision-making is a common core symptom of diseases like obsessional compulsive disorders, Tourette’s syndrome, attention deficit disorder, pathological gambling, or addiction [193]. Behavioral (cognitive) flexibility means that the animal can modify its behavior in response to modifications of the environment. Impulsivity has been defined as “actions which are poorly conceived, prematurely expressed, unduly risky or inappropriate to the situation” [194]. Different forms of impulsivity have been reported and studied [195,196]. Compulsivity refers to “actions which persist inappropriate to the situation, have no obvious relationship to the overall goal and which often results in undesirable consequences” [195]. These constructs are possibly present in invertebrates including *Aplysia* [197], *Apis* [198], or *Drosophila* [199]. However, as far as we know, the involvement of 5-HT in impulsive or compulsive responses has never been studied in those species. The research on the impulsive/compulsive dimension has been furthered in vertebrates, and more particularly in mammals with the development of numerous laboratory tests to address these dimensions [195,200,201]. The 5-HT plays an important role in the impulsive/compulsive dimension.

Feather pecking (FP) in chickens is the act of a bird plucking the feathers of other birds, which can be gentle or severe [38]. FP can include aggressiveness, fearfulness, and anxiety, but it is also proposed to display compulsive traits [202]. Deficiency in the 5-HT system can predispose birds to develop FP. Lines with high FP tendency generally have low central 5-HT and DA turnovers at young age but high turnovers in adult age. It has been reported that 5-HT_1A_R and 5-HT_1B_R agonism increases FP while D2R antagonism reduces it. Injection of S-15535 (5-HT_1A_ somatodendritic autoreceptor agonist and post-synaptic 5-HT_1A_ receptor antagonist) increased FP in chicks and this effect was associated with decreased 5-HT turnovers in the hippocampus, archistriatum, and forebrain [202]. 

In mammals, the impulsive/compulsive trait involves monoaminergic systems of the midbrain including 5-HT neurons, which are thought to shape complex interactions between prefrontal and orbital cortices, and the basal ganglia, a group of subcortical structures involved in the control of motor behavior [195,203]. Premature responses in numerous behavioral tests are often associated with a reduction of 5-HT tone [195,196]. Similarly, perseverative errors in behavioral tasks such as extinction and reversal learning paradigms in rodents and primates occur when 5-HT is depleted [204,205]. This suggests that the responses to previously rewarded stimuli that are no longer reinforced are difficult to stop in animals with low levels of 5-HT. The influence of 5-HT on impulsive/compulsive actions involves several 5-HTRs including 5-HT_2A_R and 5-HT_2C_R [129,132,206,207,208,209,210]. 

Deciphering the influence of 5-HT in the impulsive/compulsive dimension remains a complex task due to the multiple neurobiological networks involved [195,211]. When looking at the behavioral performances of a large cohort of inbred Wistar rats in six different paradigms addressing impulsivity, compulsivity, cognitive flexibility, and risk-taking, it was shown that some individuals had maladaptive responses in some but not all the tests [212]. It illustrates a broad spectrum of inter-individual differences among a cohort. The analysis of the neurochemistry of the 5-HT system and the reorganization of the data according to the performances (adaptive versus maladaptive) in a test showed that there were variations in a few, restricted brain regions for the ratio 5-HIAA/5-HT or the 5-HT content between the groups. It is noteworthy that the differences were not necessarily toward a decrease of the 5-HIAA/5-HT in the groups characterized as “impulsive” [212]. In most but not all cases, it was also associated with changes in the dopaminergic markers in the same or distal regions. 

## 12. Discussion

The participation of the 5-HT system in similar behaviors across the animal kingdom suggests that its function is conserved in most animal phyla. It can be also observed in functions that have not been covered in this review such as pain [213,214,215,216]. Moreover, in several behavioral responses, several 5-HTRs can be involved implying tight 5-HT regulatory mechanisms in responses to various stimuli. This property is interesting in order to explore deeper into the 5-HT mechanisms leading to maladaptive responses of the organisms. The transposition of this property to launch preclinical models of drug screening is rather elusive. Indeed, while the function would be conserved, the organization of the systems in interaction with the 5-HT neurons is obviously different. Moreover, the 5-HTRs are different between species, a concern that is true even between mammal species. It is still possible to express human 5-HTRs in some species like *Drosophila*, but it was still a limited value, as the 5-HTR would develop distinct pharmacological features depending on the cells expressing the 5-HTRs. 

The study of the responses of the 5-HT system is pertinent in case of the adaption of one species to its environment. Numerous ecological systems evolve due to pollution and waste drug release in the context of climatic change. It seems that antidepressants such as venlafaxine can alter the development of amphibians and fish, but also the metabolism of zebrafish as an adult. Environmental threats can induce various changes in behaviors that should be observed on 5-HT function. A loss of reproductive cycle of the mussel *Perna perna* along the Atlantic coast of Morocco has been reported, in sites nearby domestic waste or industrial pollution compared to a cleaner site. The analysis of 5-HT content in cerebroid ganglia, pedal ganglia, and gonads revealed that the temporality of the changes of 5-HT accompanying the mussels to spawning was disrupted in polluted sites. The interaction with the other monoamines dopamine and noradrenaline was completely disorganized along the reproductive cycle. Thus, it is possible that the study of the 5-HT responses in various body regions constitutes an interesting tissue marker of adaptive responses of an organism, even more, sensitive than the study of classical markers of toxicological responses. 

## 13. Conclusions

Serotonin (5-HT) is present on most animal phyla and plays adaptive roles in animal behavior and cognition both in invertebrates and vertebrates. This modulatory role thus appeared early in evolution and has been well conserved in the animal kingdom, however with species-specific effects on 5-HT targets linked to adaptation to specific environments. Thus, the neuromodulatory influence of the 5-HT system is capable of mediating opposite effects on cognition and behaviors, and it is, therefore, challenging to attribute a specific role to 5-HT in animals except for an adaptive neuromodulatory role. This global adaptive function of 5-HT results from complex interactions between neuronal outcomes depending on presynaptic or postsynaptic sites, with various and sometimes opposite effects depending on the diversity 5-HT receptors and the diversity of neurons of the networks. This diversity of actions of 5-HT is necessary for the fine-tuning of most behaviors and is the key to their adaptation to complex environments. 

## Figures and Tables

**Figure 1 ijms-21-01649-f001:**
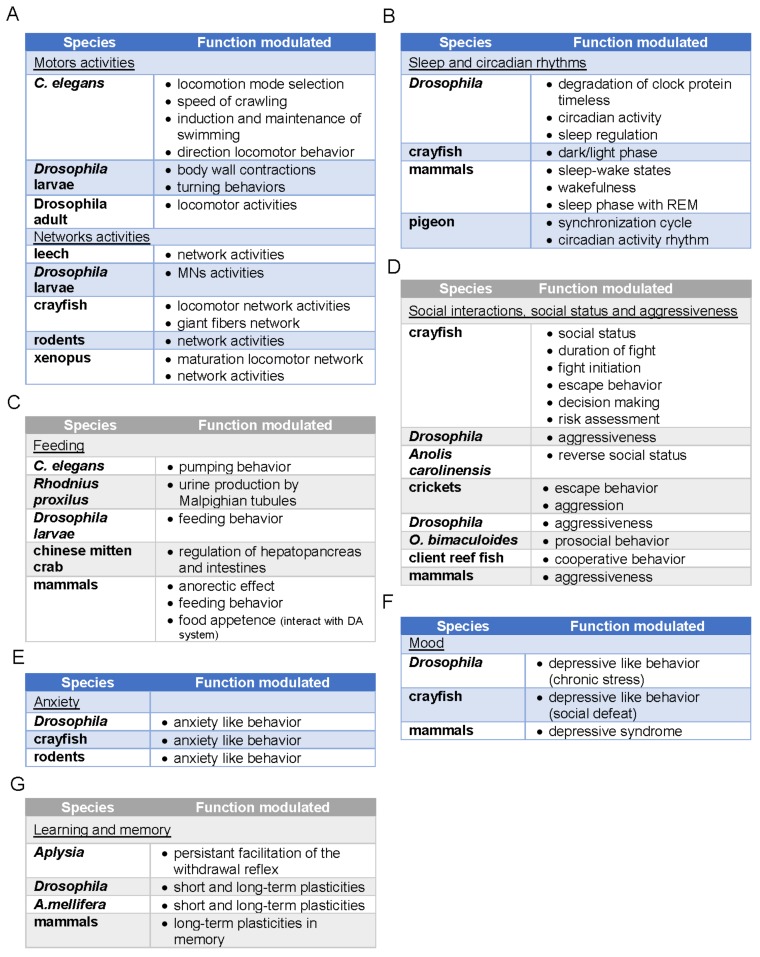
Functions modulated by 5-hydroxytryptamine (5-HT) in different species. (**A**) functions modulated related to motor activities and locomotion, (**B**) functions modulated related to sleep and circadian rhythms, (**C**) functions modulated related to sleep and circadian rhythms, (**D**) functions modulated related to social interactions, social status and aggressiveness, (**E**) functions modulated related to anxiety, (**F**) functions modulated related to mood, (**G**) functions modulated related to learning and memory.

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
