# Peer review of "Serotonin in Animal Cognition and Behavior"

_ijms, 2020, doi:10.3390/ijms21051649_

Round 1

Reviewer 1 Report

The review is a comprehensive summary of actual knowledge about the serotonergic system and its functions.

Authors analyze the different organisations and related implications in the numerous animal models studied in research.A few english typos are present through the text (e.g. raphe is repeated at line 56), but the review is generally well written and fluent.

I have minor concerns about the content:

-in the paragraph about learning and memory 5HT4 and 5HT6receptors should be cited for their role in memory modulation. See the development of 5HT4 agonists/5HT6 antagonists drug development.

5HT4 receptors have been also implicated in feeding behavior

Int J Mol Sci. 2018 Nov; 19(11): 3581. Published online 2018 Nov 13. doi: 10.3390/ijms19113581   Finally, the review would extremely benefit from a table or a figure (eventually the two), summarizing the major points and guiding the redear through the text.  

Author Response

We added in the text the minor concerns pointed by the reviewer.

-in the paragraph about learning and memory 5HT4 and 5HT6receptors should be cited for their role in memory modulation. See the development of 5HT4 agonists/5HT6 antagonists drug development.

It has been added lines 457-459.

-5HT4 receptors have been also implicated in feeding behavior

Int J Mol Sci. 2018 Nov; 19(11): 3581. Published online 2018 Nov 13. doi: 10.3390/ijms19113581  

We added this reference lines 258-260.

Finally, the review would extremely benefit from a table or a figure (eventually the two), summarizing the major points and guiding the redear through the text.  

We added a table of contents, mentionned in the introduction lines 48-50

Reviewer 2 Report

Dear editor,

the manuscript  proposed by  Baxque-Cazenave and colleagues is aimed to overview the impact of the serotonergic transmission on several ethological aspects of behavior, including motor activity, arousal, sleep, social interaction, mood learning and memory. The serotonin is an important neurotransmitter and its functioning is not deeply understood considering also its vastness. Therefore, it is of paramount importance to have updates available. I find the review very detailed and I do not find very major concerns. Just few suggestion to increase the appeal to the reader:

1) I would suggest to add  a table in which the authors could summarize the difference and multifaceted impacts of serotonin in different behaviours. This will make the manuscript more attractive and easy to follow to the readers. 

2) A little bit more emphasis on the dorsal raphe nucleus, the brain regions containing the highest amounts of serotonergic neurons would be appreciated 

3)I would be very pleased if  the author could just mention the implications of serotonin in the modulation on neuropathic pain (see De Gregorio et al. 2019 "Cannabidiol modulates serotonergic transmission and reverses both allodynia and anxiety-like behavior in a  model of neuropathic pain". PAIN, 160 (1), 136) and also in the mechanism of action of hallucinogenic compounds like LSD (see the works of Danilo De Gregorio and Gabriella gobbi, please ).

Author Response

We added minor concerns poited by the reviewer.

1) I would suggest to add  a table in which the authors could summarize the difference and multifaceted impacts of serotonin in different behaviours. This will make the manuscript more attractive and easy to follow to the readers. 

We added a table of contents, mentioned in the introduction lines 48-50 , showing the variety of 5-HT effects on different species.

2) A little bit more emphasis on the dorsal raphe nucleus, the brain regions containing the highest amounts of serotonergic neurons would be appreciated

A sentence has been added lines 61-63.

3)I would be very pleased if the author could just mention the implications of serotonin in the modulation on neuropathic pain (see De Gregorio et al. 2019 "Cannabidiol modulates serotonergic transmission and reverses both allodynia and anxiety-like behavior in a model of neuropathic pain". PAIN, 160 (1), 136) and also in the mechanism of action of hallucinogenic compounds like LSD (see the works of Danilo De Gregorio and Gabriella gobbi, please ).

The demand of the reviewer has been considered for pain with additional references (Lines 514-518). However, the mechanism of action of hallucinogenic drugs is too far from our topic.